# The Antitumor Effect of Caffeic Acid Phenethyl Ester by Downregulating Mucosa-Associated Lymphoid Tissue 1 via AR/p53/NF-κB Signaling in Prostate Carcinoma Cells

**DOI:** 10.3390/cancers14020274

**Published:** 2022-01-06

**Authors:** Kang-Shuo Chang, Ke-Hung Tsui, Shu-Yuan Hsu, Hsin-Ching Sung, Yu-Hsiang Lin, Chen-Pang Hou, Pei-Shan Yang, Chien-Lun Chen, Tsui-Hsia Feng, Horng-Heng Juang

**Affiliations:** 1Department of Anatomy, College of Medicine, Chang Gung University, Kwei-Shan, Taoyuan 33302, Taiwan; D000016684@cgu.edu.tw (K.-S.C.); hsusy@mail.cgu.edu.tw (S.-Y.H.); hcs@mail.cgu.edu.tw (H.-C.S.); 2Department of Urology, Shuang Ho Hospital, New Taipei City 235041, Taiwan; t2130@s.tmu.edu.tw; 3TMU Research Center of Urology and Kidney, Department of Medicine, College of Medicine, Taipei Medical University, Taipei 11031, Taiwan; 4Graduate Institute of Biomedical Sciences, College of Medicine, Chang Gung University, Kwei-Shan, Taoyuan 33302, Taiwan; 5Department of Urology, Chang Gung Memorial Hospital-Linkou, Kwei-Shan, Taoyuan 33302, Taiwan; laserep@mail.cgu.edu.tw (Y.-H.L.); glucose1979@cgmh.org.tw (C.-P.H.); a9307@cgmh.org.tw (P.-S.Y.); clc2679@cgmh.org.tw (C.-L.C.); 6School of Nursing, College of Medicine, Chang Gung University, Kwei-Shan, Taoyuan 33302, Taiwan; thf@mail.cgu.edu.tw

**Keywords:** MALT1, CAPE, prostate cancer, AR, NF-κB, p53, NDRG1, BTG2

## Abstract

**Simple Summary:**

The effect of caffeic acid phenethyl ester (CAPE) on prostate cancer has not been thoroughly explored. CAPE downregulated the expression of androgen receptor (AR) and mucosa-associated lymphoid tissue 1 (*MALT1)* but enhanced that of p53, thus decreasing androgen-induced activation of MALT1 and prostate-specific antigen expressions in AR-positive prostate carcinoma cells. CAPE inhibited the activity of NF-κB in p53- and AR-negative prostate carcinoma cells. Although CAPE induced the ERK/JNK/p38/AMPKα1/2 signaling pathways, pretreatment with the corresponding inhibitors of MAPK or AMPK1/2 did not inhibit the CAPE effect on MALT1 blocking. Our results reveal that CAPE blocks the expression of the *MALT1* gene to decrease the cell proliferation, invasion, and tumor growth of prostate carcinoma cells via the p53 and NF-κB signaling pathways, and they further verify that CAPE is an effective antitumor agent for human androgen-dependent and -independent prostate carcinoma cells by inhibiting MALT1 expression in vitro and in vivo.

**Abstract:**

Caffeic acid phenethyl ester (CAPE), a honeybee propolis-derived bioactive ingredient, has not been extensively elucidated regarding its effect on prostate cancer and associated mechanisms. The mucosa-associated lymphoid tissue 1 gene (*MALT1*) modulates NF-κB signal transduction in lymphoma and non-lymphoma cells. We investigated the functions and regulatory mechanisms of CAPE in relation to *MALT1* in prostate carcinoma cells. In p53- and androgen receptor (AR)-positive prostate carcinoma cells, CAPE downregulated AR and *MALT1* expression but enhanced that of p53, thus decreasing androgen-induced activation of MALT1 and prostate-specific antigen expressions. p53 downregulated the expression of MALT in prostate carcinoma cells through the putative consensus and nonconsensus p53 response elements. CAPE downregulated *MALT1* expression and thus inhibited NF-κB activity in p53- and AR-negative prostate carcinoma PC-3 cells, eventually reducing cell proliferation, invasion, and tumor growth in vitro and in vivo. CAPE induced the ERK/JNK/p38/AMPKα1/2 signaling pathways; however, pretreatment with the corresponding inhibitors of MAPK or AMPK1/2 did not inhibit the CAPE effect on MALT1 blocking in PC-3 cells. Our findings verify that CAPE is an effective antitumor agent for human androgen-dependent and -independent prostate carcinoma cells in vitro and in vivo through the inhibition of MALT1 expression via the AR/p53/NF-κB signaling pathways.

## 1. Introduction

Propolis (bee glue), produced naturally by honeybees, has been widely used in traditional medicine because of its antimicrobial, anti-inflammatory, and blood pressure- and cholesterol-lowering effects [1]. Caffeic acid phenethyl ester (CAPE), the main bioactive component of propolis, has antioxidant, antiviral, antibacterial, anti-inflammatory, and antitumor properties [1,2,3,4,5]. Moreover, CAPE specifically inhibits nuclear transcription factor κB (NF-κB) by preventing the translocation of NF-κB subunits to the nucleus [6]. According to earlier reports, the inhibitory effects of CAPE on the tumor growth of several cancer cells, including prostate cancer cells, have been evaluated in in vitro or in vivo animal studies [7,8,9,10,11,12,13]. Our previous studies also demonstrated that CAPE represses the growth of oral squamous carcinoma cells (OSCCs), nasopharyngeal cancer (NPC), and bladder carcinoma cells (BCCs) by activating the MAPK signaling pathway to induce the expression of N-myc downstream regulated 1 (NDRG1) [14,15,16].

Mucosa-associated lymphoid tissue protein 1 (MALT1), a lymphoma oncogene, functions as an adaptor protein and paracaspase in the regulation of antigen receptor-mediated signaling in the NF-κB pathway [17,18,19]. MALT1 mediates Lys63-linked polyubiquitination to activate the IκB kinase (IKK) complex, which is essential for inhibiting nuclear factor kappa B (IκB) phosphorylation and NF-κB nuclear translocation [20,21,22]. MALT1 also plays distinct roles in the progression of solid nonlymphoid tumors, including breast cancer, lung cancer, melanoma, and cholangiocarcinoma [23,24,25,26]. Our recent study further revealed that MALT1 activates the IKK complex to induce nuclear NF-κB accumulation in prostate carcinoma cells [27]. However, in the human prostate, whether CAPE is a MALT1 expression modulator, and the regulatory mechanisms of CAPE on MALT1, remain unclear, even though studies have indicated that CAPE downregulates NF-κB by inhibiting the translocation of NF-κB subunits into the nucleus [6,12]. This study ascertained that CAPE can suppress MALT1 expression via upregulation of p53, anti-NF-κB, and antiandrogen activations, as well as attenuating the growth, proliferation, and invasion of prostate carcinoma cells both in vitro and in vivo.

## 2. Materials and Methods

### 2.1. Cell Lines and Cell Culture

The LNCaP, 22Rv1, PC-3, and DU145 cell lines were obtained from the Bioresource Collection and Research Center (Hsinchu, Taiwan) and maintained in RPMI 1640 medium with 10% fetal calf serum (FCS), as described previously [28]. 22Rv1 cells, an *AR*-positive but androgen-independent cell line, were derived from a xenograft and serially propagated in mice after castration-induced regression and relapse of the parental, androgen-dependent CWR22 xenograft [29]. The FCS was purchased from HyClone Laboratories (Logan, UT, USA), RPMI 1640 medium from Life Technologies (Gaithersburg, MD, USA), Matrigel from BD Biosciences (San Jose, CA, USA), methyltrienolone (R1881) from NEN Life Sciences (Boston, MA, USA), ERK inhibitor (PD0325901), p38 inhibitor (SB202190), AMPK inhibitor (dorsomorphin), and camptothecin from Sigma (St. Louis, MO, USA), CAPE from Selleckchem (Houston, TX, USA), and JNK inhibitor II (SP600125) from Merck Millipore (Burlington, MA, USA). For the androgen treatment study, the cells were incubated in phenol red-free medium (RPMI-PRF) containing 5% charcoal-dextran FCS (CD-FCS) to deplete endogenous steroid hormones.

### 2.2. Cell Proliferation Assays

Cells (2.5 × 10^5^ to 5 × 10^5^) were cultured in serum-free medium for 24 h. After incubation in 10% serum for another 48 h, the cells were incubated with 10 μM 5-ethynyl-2′-deoxyuridine (EdU) for 2 h. The cells were subsequently collected using trypsin-EDTA, centrifuged at 500× *g* for 10 min and analyzed using Click-iT EdU Flow Cytometry Assay Kits (Thermo Fisher Scientific, Waltham, MA, USA), as described previously [30].

### 2.3. Expression Vector Constructs and Stable Transfection

The p53 expression vectors were constructed as described previously [31]. The expression vectors were introduced into prostate carcinoma PC-3 cells by electroporation using the ECM 830 (BTX, Holliston, MA, USA) set at a single pulse setting, 70 ms, and 180 V. The mock transfection cells were transfected with control pcDNA3 expression vectors (Invitrogen, Carlsbad, CA, USA) and clonally selected in the same manner as gene overexpression cells.

### 2.4. Nuclear and Cytoplasmic Extraction Assay

Cells (2 × 10^6^) were seeded in a T75 flask and incubated for 1–2 d. After incubation, cells were harvested with trypsin-EDTA and then centrifuged at 500× *g* for 5 min. Cell pellets were washed twice with PBS and pelleted again by centrifugation at 500× *g* for 5 min. Nuclear and cytoplasmic fractions were separated using NE-PER^TM^ nuclear and cytoplasmic extraction kits (Thermo Fisher Scientific), as described previously [32].

### 2.5. Immunoblot Assay

The protein lysates (20 or 40 μg) were separated on 10% or 12% sodium dodecyl sulfate-polyacrylamide gels. The blotting membranes were probed with β-actin antiserum (MAB1501, Merck Millipore), GAPDH (6C5, Santa Cruz Biotechnology, Dallas, TX, USA), MALT1 (EP603Y, Abcam, Cambridge, MA, USA), p53 (DO-1, Santa Cruz Biotechnology), NF-κB p50 (06-886, Merck Millipore), NF-κB p65 (06-418, Merck Millipore), Lamin B1 (D9V6H, Cell Signaling Technology, Danvers, MA, USA), IκB-α (#9242, Cell Signaling Technology), p-IκB-α (#2859, Cell Signaling Technology), p44/42 MAPK (Erk1/2; #4695, Cell Signaling Technology), phospho-p44/42 MAPK (Erk1/2; #9101, Cell Signaling Technology), SAPK/JNK (#9258, Cell Signaling Technology), phospho-SAPK/JNK (#4668, Cell Signaling Technology), p38 MAPK (#8690, Cell Signaling Technology), phospho-p38 MAPK (#9211, Cell Signaling Technology), AMPKα1/2 (#5831, Cell Signaling Technology), phospho-AMPKα1/2 (#40H9, Cell Signaling Technology), NDRG1 (42-6200, Thermo Fisher Scientific), AR (N-20, Santa Cruz Biotechnology), prostate-specific antigen (PSA; A0562, Dako Denmark A/S, Glostrup, Denmark), maspin (554292, BD Bioscience), and BTG2 (GWB-D54FE7, GenWay Biotech, San Diego, CA, USA). Band intensities were detected using the Western lightning plus-ECL detection system (PerkinElmer, Waltham, MA, USA), recorded using the LuminoGraph II (Atto Corporation, Tokyo, Japan), and analyzed using the GeneTool Program of ChemiGenius (Syngene, Cambridge, UK).

### 2.6. Real-Time Reverse Transcriptase-Polymerase Chain Reaction

Total RNA from cells was isolated using TRIzol reagent (Invitrogen). cDNA was synthesized using the SuperScript III (Invitrogen) system, and real-time polymerase chain reaction (qPCR) was performed using a CFX Connect Real-Time PCR system (Bio-Rad Laboratories, Foster City, CA, USA), as described previously [33]. qPCR FAM dye-labeled TaqMan MGB probes for β-actin (Hs01060665_g1), *BTG2* (Hs00198887_m1), *MALT1* (Hs01120052_m1), *NDRG1* (Hs00608387_m1), and *p53* (Hs01034249_m1) were purchased from Applied Biosystems (Foster City, CA, USA). The mean cycle threshold (C_t_) values for target genes were normalized against the β-actin control probe to calculate ΔC_t_ values.

### 2.7. Enzyme-Linked Immunosorbent Assay

Cells were incubated with 0.5 mL of different concentrations of drugs as indicated in RPMI-PRF medium with 5% CD-FCS in a 6-well plate (2 × 10^5^ cells/well) for 24 h. Following incubation, the culture media from each well were collected for PSA assay. Cell pellets were washed twice with ice-cold PBS and then dissolved in 500 μL of PBS. After sonication for 10 s, the cell extracts were centrifuged at 13,000× *g* for 20 min. The PSA levels in 20 μL of the culture media were measured using PSA enzyme-linked immunosorbent assay (ELISA), as described previously [34]. The PSA level in each sample was adjusted by the protein concentrations in the whole cell extract, which were measured using a BCA protein assay kit.

### 2.8. Reporter Vector Constructs

The reporter vectors, pbGL3-PSABHE (−4801 to −3933 and −589 to +41), containing the androgen response element (ARE) and 5′-flanking promoter regions of the human PSA genes, were cloned, as described previously [35]. The 5′-DNA fragment (−1 to −6313) of *MALT1*, according to the sequence from GenBank (AP005018.1), was synthesized by Invitrogen. The human *MALT1* reporter vector was constructed by cloning the DNA fragment into the pGL3-Basic reporter vector (pbGL3) (Promega Biosciences, Madison, WI, USA) with *Hind III* sites, as described previously [27].

### 2.9. Transient Transfection and Reporter Assay

PC-3 cells were transiently transfected using the X-tremeGENE HP DNA transfection reagent (Roche Diagnostics, Mannheim, Germany), and the LNCaP cells were transfected using TransFast transfection reagent (Promega Biosciences). The luciferase activity was determined in a relative light unit using the synergy H1 microplate reader (BioTek Instruments, Beijing, China). Each sample was adjusted per the protein concentrations in the whole cell extract [28].

### 2.10. Xenograft Animal Model

The current animal studies were performed in accordance with the US National Institutes of Health Guide for the Care and Use of Laboratory Animals and approved by the Chang Gung University Animal Research Committee (CGU106-157). We obtained 4-week-old male nude mice (BALB/cAnN-Foxn1) from the animal center of the National Science Council, Taipei, Taiwan. PC-3 cells were detached through treatment with Gibco Versene solution (Life Technologies) and washed with RPMI1640 medium containing 10% FBS. The mice were anesthetized intraperitoneally with a mixture of 1 mL of Zoletil 50, 0.1 mL of xylazine, and 3.9 mL of PBS (0.06 mL/10 g per mouse) and then administered 3 × 10^6^ cells in 100 μL of PBS subcutaneously into the loose skin over the interscapular area. When the xenograft tumors grew to a size of 50 mm^3^, the mice were randomly divided into 2 groups. CAPE (10 mg/kg/d) was dissolved in DMSO and intraperitoneally injected once per day for 5 d per week, as modified from a previous study [15]. The control group was treated with vehicle (0.01% DMSO in PBS). Tumor growth was measured using vernier calipers every 2–3 d. Tumor volume was determined using the following formula: (length × width^2^)/2. Mouse body weight was measured 3 times weekly. The mice were sacrificed, and the tumors of xenograft animals were collected and digested with protein lysis buffer or Trizol reagent for further analysis of the mRNA or protein expression of target genes.

### 2.11. Matrigel Invasion Assay

The invasion ability of the cells was determined through an in vitro Matrigel invasion assay, as described previously [28]. Cells that migrated to the other side of the transmembrane were fixed with 4% paraformaldehyde and stained with 0.1% crystal violet solution for 10 min. The number of cells that invaded the Matrigel was recorded and photographed microscopically (BX43, Olympus, Tokyo, Japan).

### 2.12. NF-κB (p65) Transcription Factor Binding Assay

The NF-κB (p65) binding activity was performed using an NF-κB (p65) Transcription Factor Assay Kit (Item No. 10007889, Cayman Chemical, Ann Arbor, MI, USA), as described previously [27]. Briefly, the nuclear extracts were incubated with consensus dsDNA sequences in each well overnight at 4 °C. The samples were then incubated with p65 primary antibody for 1 h, followed by goat anti-rabbit HRP conjugate incubation. The p65 binding activity was measured at 450 nm using the synergy H1 microplate reader (BioTek Instruments) after treatment with Transcription Factor Developing Solution (Cayman Chemical).

### 2.13. Statistical Analysis

The results are expressed as means ± standard errors (SEs). Statistical significance was determined using the *p* value (*p* < 0.05 or *p* < 0.01) through Student’s *t* test and one-way analysis of variance (ANOVA) on SigmaStat for Windows (version 2.03; SPSS, Chicago, IL, USA). Multiple comparisons were conducted using ANOVA with Tukey’s post hoc test.

## 3. Results

### 3.1. CAPE Attenuates Activation of Androgen in MALT1 Expression in Androgen-Positive Prostate Carcinoma Cells

We investigated the effect of CAPE on *MALT1* expression in prostate carcinoma cells. Immunoblot assays revealed that CAPE treatments downregulated MALT1, PSA, and AR expressions and upregulated p53 and NDRG1 expressions in prostate carcinoma LNCaP cells (Figure 1A and Appendix A). Furthermore, CAPE attenuated the activation of an androgen agonist (R1881) in MALT1, AR, and PSA expressions in LNCaP cells (Figure 1B) or 22Rv1 cells (Figure 1C). The results of quantitative analyses from three independent experiments are presented at the bottom of each figure. Reporter assays using the PSA reporter vector with a specific human PSA promoter and the androgen response element revealed that R1881 induced PSA expression, and that CAPE blocked the activation of R1881 in the PSA reporter activity (Figure 1D). Similar results were found with ELISA, indicating that CAPE not only attenuated PSA secretion but also blocked the activation of R1881 in PSA secretion from LNCaP cells (Figure 1E).

### 3.2. Tumor Suppressor p53 Downregulated MALT1 Expression in Prostate Carcinoma Cells

Because CAPE upregulated p53 expression in p53 wild-type prostate carcinoma LNCaP cells, as shown in Figure 1A, we determined whether p53 affects *MALT1* expression in prostate carcinoma cells. Immunoblot assays and quantitative analyses indicated that transient p53 overexpression in p53-null PC-3 cells upregulated p53 and downstream targets of antitumor genes (NDRG1 and BTG2) but downregulated MALT1 protein levels (Figure 2A and Appendix A). RT-qPCR assays revealed that p53 downregulated *MALT1* expression but upregulated *NDRG1* and *B**TG2* in transient p53-overexpressed PC-3 (PC3-p53) cells compared with mock-transfected PC-3 (PC3-DNA) cells (Figure 2B). In addition, immunoblot assays and quantitative analyses revealed that camptothecin (Cpt), a p53 inducer, induced p53 but inhibited MALT1 and PSA expression in p53 wild-type prostate carcinoma LNCaP cells dose dependently (Figure 2C). Furthermore, the results of the 5′-deletion assays demonstrate that transient p53 overexpression downregulated MALT1 promoter activity, mostly depending on the DNA region from −3048 to 6313; however, the DNA fragment of −1 to −559 might affect MALT1 reporter activity as well (Figure 2D).

### 3.3. CAPE Blocks MALT1 Gene Expression to Downregulate NF-κB Activation in Androgen-Negative Prostate Carcinoma Cells

Since we demonstrated that CAPE attenuated *MALT1* gene expression by inhibiting AR and upregulating p53 expression in both AR-positive LNCaP and 22Rv1 cells, we continued to determine whether CAPE modulates MALT1 expression in PC-3 (AR-null, p53-null) and DU145 (AR-null, p53-mutant) cells. The immunoblot assays with quantitative analyses (Figure 3A and Appendix A) and the RT-qPCR (Figure 3B) assays revealed that CAPE treatment upregulated NDRG1 but downregulated MALT1 expression dose dependently. The immunoblot assays confirmed that cytoplasmic and nuclear fractions from CAPE-treated PC-3 cells downregulated MALT1 expression and IκBα phosphorylation but upregulated IκBα expression in cytoplasmic fractions, thereby decreasing nuclear p65 and p50 protein levels (Figure 3C). In addition, the results of the NF-κB transcription factor binding assay reveal that CAPE treatments blocked NF-κB activity in PC-3 cells (Figure 3D). Further immunoblot assays (Figure 3E, left) and quantitative analyses (Figure 3E, right) demonstrated similar trends of MALT1 and NDRG1 expression after CAPE treatment in DU145 cells. Reporter assays with 5′-deletion assays revealed that CAPE treatments in PC-3 cells downregulated the reporter activity of the MALT1 reporter vector, which is dependent on the elements located on the 5′-flanking region (−1 to −559) of human *MALT1* (Figure 3F).

### 3.4. CAPE Downregulates PC-3 Cell Proliferation and Invasion

The percentage of positive cells with 5′-ethynyl-2′deoxyuridine (EdU) staining was decreased in the CAPE-treated PC-3 cells, as determined by flow cytometry. Treatment of 30 μM CAPE blocked approximately 43% of cell proliferation compared to the vehicle-treated PC-3 cells (Figure 4A). Matrigel invasion assays revealed that CAPE treatments blocked the cell invasion ability in PC-3 cells. The invasion capacity was downregulated by 64% and 91% in treatments of 10 μM and 30 μM CAPE, respectively, when compared with the vehicle-treated PC-3 cells (Figure 4B).

### 3.5. CAPE Induces Phosphorylation of ERK, p38, JNK, and AMPKα1/2 in PC-3 Cells

To assess the signaling pathways involved in CAPE treatments in prostate carcinoma cells, we treated PC-3 cells with CAPE to determine the activities of ERK, p38, JNK, and AMPKα1/2. The results of the time course immunoblot assays reveal that CAPE induced the phosphorylation of ERK, JNK, p38, and AMPKα1/2 within 15 to 30 min in PC-3 cells (Figure 5A and Appendix A). Other immunoblot assays indicated that CAPE treatments induced phosphorylation of ERK, JNK, p38, and AMPKα1/2 dose dependently at 30 min (Figure 5B). The results of quantitative analyses from three independent experiments are presented in Figure 5C–J.

### 3.6. CAPE Induced Expression of MALT1, NDRG1, and Maspin via Different Signaling Pathways

To explore whether CAPE-increased expressions of MALT1, NDRG1, and maspin in PC-3 cells were through the MAPK and AMPK signaling pathways, PC-3 cells were pretreated with inhibitors of MAPK elements or AMPK for 1 h before exposure to CAPE for 16 h. Immunoblot assays demonstrated that CAPE (30 μM) increased ERK, p38, and JNK activations; however, the expressions of p-ERK (Figure 6A and Appendix A), p-JNK (Figure 6C), p-p38 (Figure 6E), and p-AMPKα1/2 (Figure 6G) decreased after pretreatment with the corresponding inhibitors. MALT1, NDRG1, and maspin protein levels in PC-3 cells after CAPE treatments with (+) or without (−) pretreatment with PD0325901 (Figure 6B), SP600125 (Figure 6D right), SB202190 (Figure 6F), or dorsomorphin (Figure 6H) indicated that each inhibitor had different effects on MALT1, NDRG1, and maspin expressions. CAPE upregulated NDRG1 and maspin protein levels and downregulated MALT1 expression; however, cells pretreated with the ERK inhibitor (PD0325901), JNK inhibitor (SP600125), p38 inhibitor (SB202190), or AMPK inhibitor (dorsomorphin) decreased CAPE-induced upregulation of maspin expression but did not significantly affect NDRG1 expression. Notably, none of the inhibitors blocked CAPE’s effect on MALT1 expression.

### 3.7. CAPE Inhibits Tumorigenesis of PC-3 Cells

To evaluate the growth inhibitory effect of CAPE on prostate carcinoma cells in vivo, PC-3 cells were xenografted. At day 18, solid tumors were approximately 50 mm^3^ in volume, and xenografted mice were divided randomly into two groups (*n* = 6). CAPE (10 mg/kg/d) or vehicle (0.01% DMSO in PBS) was administered intraperitoneally once per day for 5 d per week after solid tumors were established. After 18 d of treatment, the tumor size and weight in CAPE-treated mice were 57% lower (169.95 ± 59.92 vs. 393.03 ± 54.98 mm^3^) and 53% lower (0.1795 ± 0.0774 vs. 0.3831 ± 0.0537 g) than those in vehicle-treated mice, respectively (Figure 7A,B). The average body weight of CAPE-treated mice (24.23 ± 0.37 g) was not significantly different from that of the vehicle-treated mice (22.58 ± 0.78 g, Figure 7C). Furthermore, CAPE-treated xenografted tumors had increased NDRG1 expression and decreased MALT1 mRNA and protein expression, as determined using immunoblot (Figure 7D,E and Appendix A) and RT-qPCR assays (Figure 7F). Taken together, the findings indicate that CAPE treatment blocked PC-3 cell growth both in vitro and in vivo.

## 4. Discussion

CAPE, the main bioactive component of propolis, specifically inhibits NF-κB by preventing the translocation of NF-κB subunits to the nucleus [6]. CAPE has been reported as a potential regulator of oncogenic molecular pathways to inhibit cell proliferation in vitro or tumor growth in vivo in animal models of various carcinoma cells including melanoma, cholangiocarcinoma, breast, lung, colon, pancreas, and prostate carcinoma cells [7,8,9,10,11,12,13]. Our previous studies also confirmed its promising anticancer effects in OSCCs, NPC, and BCCs in vitro or in vivo [14,15,16]. An earlier report indicated that the physiological concentration of CAPE in human serum is 17 μM [36]. The dosage of CAPE treatments varies from 0 to 100 μM in studies of different cell types [37]. In the present study, we found that CAPE treatment at dosages of 10–30 μM can effectively inhibit the proliferation and invasion ability of prostate carcinoma PC-3 cells, consistent with a previous study [10]. Immunoblot and RT-qPCR assays from this study revealed that CAPE induced NDRG1 expression but inhibited MALT1 expression in PC-3 cells in vitro and in vivo (Figure 3, Figure 6, and Figure 7). The CAPE-induced upregulation of NDRG1 expression is consistent with our previous studies on OSCCs, NPC, and BCCs [14,15,16].

Our in vivo animal study indicated that 10 mg/kg of CAPE treatment repressed tumorigenesis in a PC-3 xenograft model without affecting the body weight of mice, similar to studies on OSCCs and BCCs [14,16]. Previous studies indicated that intraperitoneal injection of 10 mg/kg of CAPE did not show any toxicity, while higher doses (e.g., 20–30 mg/kg) may cause mild dose-dependent toxicity in the liver and kidney of mice [10,11]. Although, at present, clinical trials have not found sufficient evidence for CAPE’s effects on human cancers, there was a clinical trial record (NCT02050344) which indicated that taking up to 20 mg/kg of CAPE orally only caused a minor side effect in a few participants. Further clinical studies of the application of CAPE or its analogs against prostate cancer are warranted.

Serum PSA, a well-known marker of prostate cancer detection [38], is induced by androgens [34,35]. The present study verified that CAPE downregulated AR, MALT1, and PSA expressions and upregulated p53 expression. In agreement with a previous study, CAPE treatments significantly increased the abundance of the p53 protein in p53 wild-type prostate carcinoma cells in this study [10]. Moreover, CAPE attenuated androgen-induced PSA and MALT1 expressions in AR-positive prostate carcinoma cells (LNCaP and 22Rv1). These results are consistent with a study indicating that CAPE suppressed AR signaling [39]. Prior studies have shown that AR variants such as ARV7 modulated the NF-κB activity in prostate carcinoma cells [40,41]. Whether CAPE modulates the NF-κB activity via the modulation of AR variant expression is still unknown and worthy of further investigation. In addition, comparison of CAPE with antiandrogens or androgen antagonists in clinical studies is warranted. The mechanisms underlying the p53-induced repression of PSA expression have been previously elucidated [42,43]. An earlier study indicated that PSA expression was upregulated by NF-κB activity in androgen-independent prostate carcinoma cells [44]. Consistent with this study, our results indicate that CAPE downregulated MALT1, which downregulated NF-κB activity (Figure 3), and also that CAPE attenuated androgen-induced PSA via the AR, MALT1/NF-κB, or p53 signaling pathway in androgen-dependent LNCaP and 22Rv1 cells (Figure 1).

The present study is the first to identify MALT1 as a p53-downreglated oncogene. As shown in Figure 2, p53 upregulated the antitumor genes NDRG1 and BTG2, consistent with previous studies [31,33,45], but downregulated MALT1 expression in human prostate carcinoma cells, which may lead to an antitumor effect. Moreover, camptothecin-induced endogenous p53 downregulated PSA expression in LNCaP cells, which is in agreement with our previous results [43]. p53 is a crucial tumor suppressor and acts as a transcription factor that recruits coactivators and corepressors, and it binds to the sequence of the p53 consensus response element (5′-RRRCWWGYYYN(0-13)RRRCWWGYYY) to either promote or inhibit the transcription of different target genes [46]. In the present study, p53 decreased MALT1 reporter activity mostly depending on the DNA 5′-flanking region in human MALT1 from −3048 to −6313. A putative tumor suppressor p53 binding site (AAGCTAGTTTTGCATCTAGGAAGCAACTTC-3′) was found in silico on the MALT1 promoter region at the −3668 site. However, reporter assays also revealed that the effect of p53 on MALT1 may depend on the 5′-flanking region (−559 to −1) of the MALT1 promoter, which does not contain the putative consensus p53 response element. An earlier study revealed that only 5% of p53 target genes contain a p53 consensus site using an unbiased global chromatin immunoprecipitation assay [47]. Future studies elucidating the precise mechanisms of p53 on MALT1 expression are warranted.

Additionally, our results demonstrate that CAPE inhibited MALT1 expression in AR-null PC-3 and DU145 cells. Unlike LNCaP cells, both PC-3 and DU145 cells have constitutively active NF-κB signaling [48,49]. Our recent study verified that MALT1 is an NF-κB-upregulated oncogene, and a positive feedback loop was found in prostate carcinoma cells [27]. In alignment with CAPE, an NF-κB signaling inhibitor, the current study proves that CAPE blocked NF-κB activity by downregulating MALT1 expression in PC-3 cells (Figure 3). Moreover, CAPE inhibited not only AR activity but also proliferation and invasion in AR-positive and AR-negative prostate carcinoma cells, consistent with previous reports [11,39]. Our xenograft animal study illustrates that CAPE treatment effectively attenuated tumor growth in vivo by downregulating MALT1 and upregulating NDRG1 expression in PC-3 cells (Figure 7). CAPE treatment upregulated NDRG1 expression in PC-3 cells in vivo, which is consistent with previous studies on OSCCs and BCCs [14,16].

The MAPK pathways triggered oncogenic and tumor suppressor roles which contributed to several diverse cellular activities including mitosis, metabolism, motility, survival, apoptosis, and differentiation [50]. Our data indicate that CAPE activated ERK, JNK, and p38 signaling in PC-3 cells (Figure 5 and Figure 6), consistent with studies on OSCCs, NPC, and BCCs which demonstrated that CAPE influences NDRG1 via the MAPK signaling pathway [14,15,16]. However, PC-3 cells pretreated with an ERK, JNK, p38, or AMPK inhibitor lowered the CAPE-induced increase in maspin expression but did not significantly affect the CAPE-induced increase in NDRG1 expression. The effect of CAPE treatment on different cell types in vitro or in vivo involving other signaling pathways needs further investigation. Maspin is a CAPE-induced antitumor gene in BCCs [16,30]. One recent study using microarray analysis and NGS (next-generation sequencing) in colon cells after treatment with CAPE found several novel genes modulated by CAPE treatment [9]; therefore, it is worth continuously exploring CAPE-modulated genes in prostate carcinoma cells. Notably, this is the study firstly to reveal that CAPE induced AMPKα1/2 in prostate carcinoma cells. Interestingly, CAPE downregulated MALT1 expression in PC-3 cells, and this downregulation was also not blocked by any of the corresponding inhibitors; thus, CAPE may not affect MALT1 expression via the MAPK and AMPK signaling pathways in prostate carcinoma cells. Taken together, our results suggest that CAPE influences MALT1 expression not via the ERK, JNK, p38, and AMPKα1/2 signaling pathways but via the NF-κB and p53 signaling pathways.

## 5. Conclusions

In conclusion, CAPE can effectively and safely repress the growth and invasiveness of prostate carcinoma cells both in vitro and in vivo; this mechanism potentially leads to therapeutic effects on prostate cancer. CAPE attenuated androgen-induced PSA via the AR, MALT1/NF-κB, or p53 signaling pathway in androgen-dependent and androgen-independent prostate carcinoma cells. Our findings reveal that *MALT1*, *AR*, *p53*, *NDRG1*, and *MASPIN* are CAPE-modulated genes in prostate carcinoma cells and responsible for CAPE-mediated cell growth inhibition. CAPE activated ERK, JNK, p38, and AMPKα1/2; however, it did not affect MALT1 via these signaling pathways but perhaps via the NF-κB and p53 signaling pathways in prostate carcinoma cells.

## Figures and Tables

**Figure 1 cancers-14-00274-f001:**
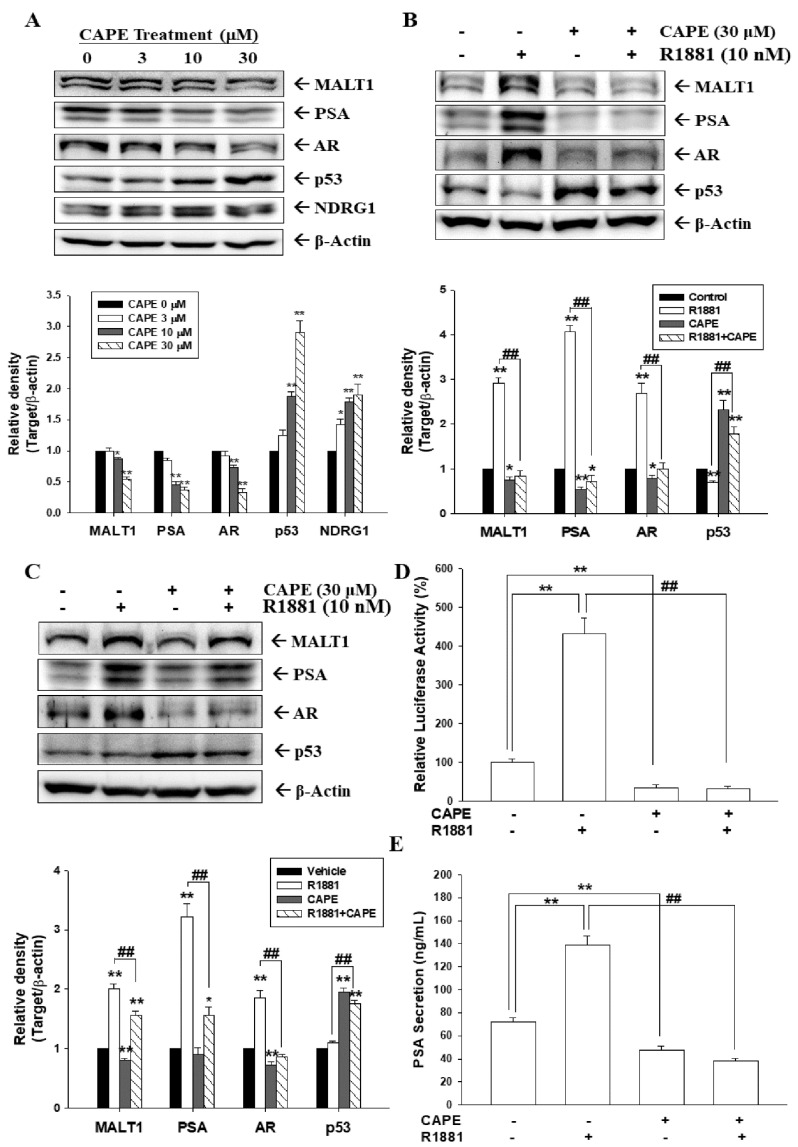
CAPE downregulates MALT1 expression in AR−positive prostate carcinoma cells. (**A**) LNCaP cells were treated with various concentrations of CAPE for 24 h. Cells were lysed, and MALT1, PSA, AR, p53, NDRG1, and β-actin expressions were determined using immunoblotting. LNCaP (**B**) and 22Rv1 (**C**) cells were treated with/without 30 μM CAPE and 10 nM R1881 as indicated for 24 h. Cells were lysed, and MALT1, PSA, AR, p53, and β-actin expressions were determined using immunoblotting. The quantitative data are expressed as the intensity of protein bands of the target proteins/β-actin relative to the control vehicle-treated group. (**D**) The reporter activity of the PSA reporter vector cotreated with 30 μM CAPE and/or 10 nM R1881. (**E**) Relative PSA protein secretion in LNCaP cells after treatment with/without R1881 and CAPE for 24 h. Data are expressed as mean percentage ± SE (*n* = 6) compared with the vehicle-treated group (* *p* < 0.05, ** *p* < 0.01) or the R1881-treated group (## *p* < 0.01).

**Figure 2 cancers-14-00274-f002:**
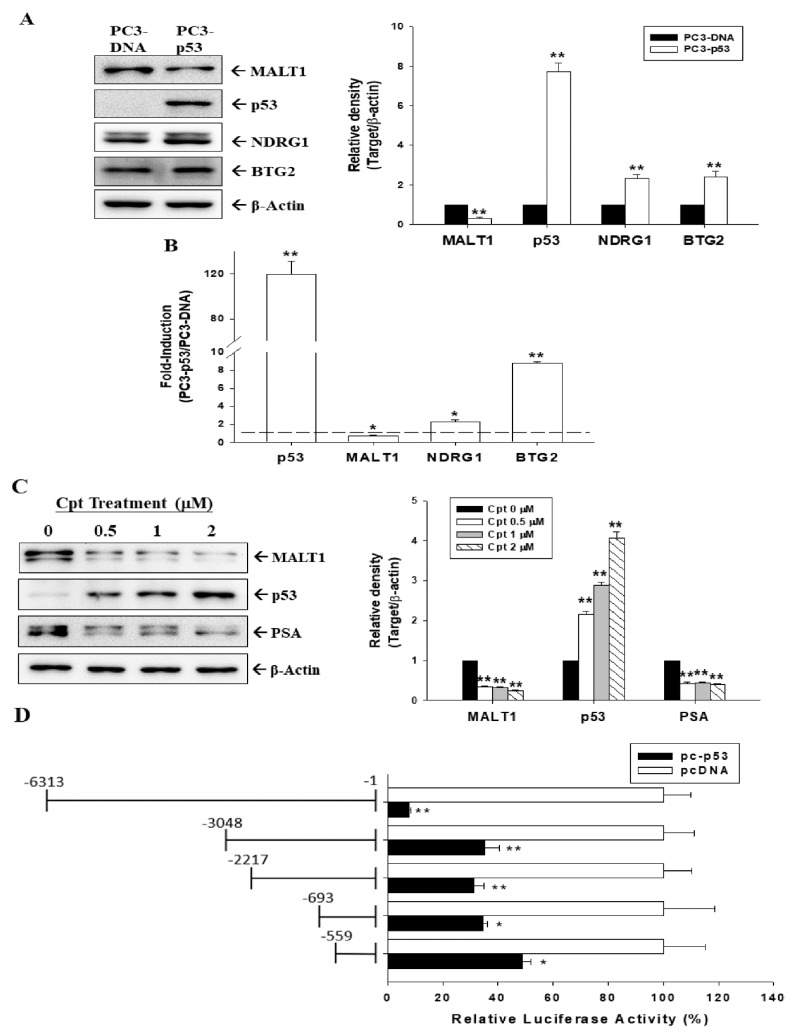
Modulation of p53 in MALT1 expression in prostate carcinoma cells. (**A**) The mock-transfected PC-3 (PC3-DNA) and p53-overexpressed PC-3 (PC3-p53) cells were lysed, and MALT1, p53, NDRG1, BTG2, and β-actin expressions were determined using immunoblotting. (**B**) Relative fold induction mRNA levels of p53, MALT1, NDRG1, and BTG2 in PC3-p53 cells compared with PC3-DNA cells determined using RT-qPCR assays. (**C**) LNCaP cells were treated with camptothecin (0–2 μM) for 16 h. The cells were lysed, and MALT1, p53, PSA, and β-actin expressions were determined using immunoblotting. The quantitative data are expressed as the intensity of protein bands of the target proteins/β-actin relative to the control vehicle-treated group. (**D**) Relative luciferase activity of PC-3 cells cotransfected with nested deletion constructs of the human MALT1 reporter vectors cotransfected with (black bars) or without (white bars) p53 expression vectors. Data are presented as mean percentage (±SE, *n* = 6) compared with the control mock-transfected group. * *p* < 0.05, ** *p* < 0.01.

**Figure 3 cancers-14-00274-f003:**
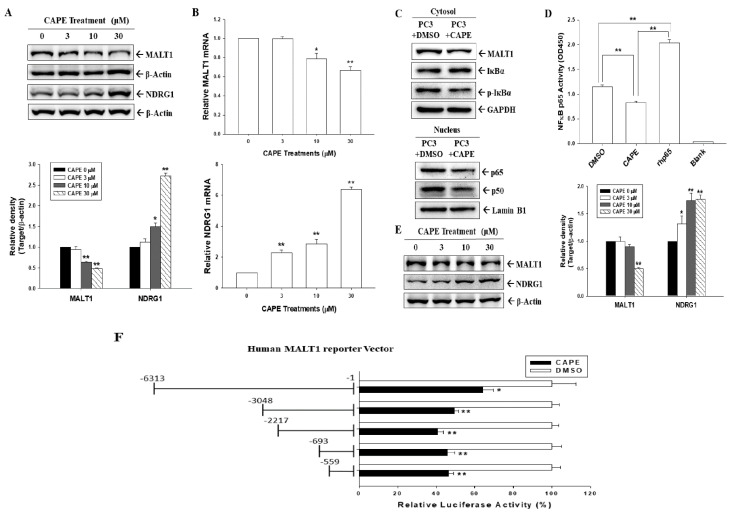
Effects of CAPE on NF-κB activation in AR−negative PC-3 cells. (**A**) The PC-3 cells were treated with various concentrations of CAPE for 24 h. Cells were lysed, and MALT1, NDRG1, and β-actin expressions were determined using immunoblotting. The quantitative data are expressed as the intensity of protein bands of the target proteins/β-actin relative to the control vehicle-treated group. (**B**) The PC-3 cells were treated with various concentrations of CAPE for 24 h. The mRNA levels of MALT1 (top) and NDRG1 (bottom) were determined using RT-qPCR assays (±SE, *n* = 3). (**C**) Expressions of MALT1, IκBα, phospho-IκBα, p65, p50, Lamin B1, and GAPDH in PC-3 cells treated with/without CAPE after separation of nuclear and cytoplasmic fractions determined by immunoblot assays. (**D**) The NF-κB (p65) binding activity in CAPE-treated PC-3 cells. (**E**) The DU145 cells were treated with various concentrations of CAPE as indicated for 24 h. Cells were lysed, and MALT1, NDRG1, and β-actin expressions were determined using immunoblotting (left). The quantitative data are expressed as the intensity of protein bands of the target proteins/β-actin relative to the control vehicle-treated group (right). (**F**) Relative luciferase activity of PC-3 cells transfected with nested deletion constructs of the human MALT1 reporter vectors and treated with (black bars) or without (white bars) 30 μM CAPE for 24 h. Data are presented as mean percentage of reporter activity (±SE, *n* = 6) compared with the vehicle-treated group. * *p* < 0.05, ** *p* < 0.01.

**Figure 4 cancers-14-00274-f004:**
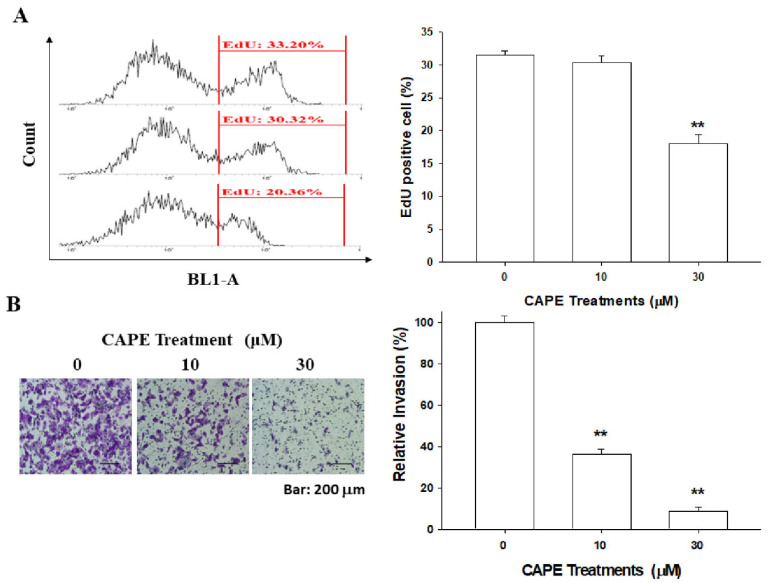
Effects of CAPE on cell proliferation and invasion in AR−negative PC-3 cells. (**A**) The proliferation ability of PC-3 cells after treatment with various concentrations of CAPE as indicated for 24 h was determined through flow cytometry by using the Click-iT EdU flow cytometry kit (±SE, *n* = 4). (**B**) The invasion ability of PC-3 cells after treatment with various concentrations of CAPE as indicated for 24 h was determined through in vitro Matrigel invasion assays (±SE, *n* = 3). The bar line represents 200 μm. Data are presented as mean percentage of reporter activity (±SE, *n* = 6) compared with the vehicle-treated group. ** *p* < 0.01.

**Figure 5 cancers-14-00274-f005:**
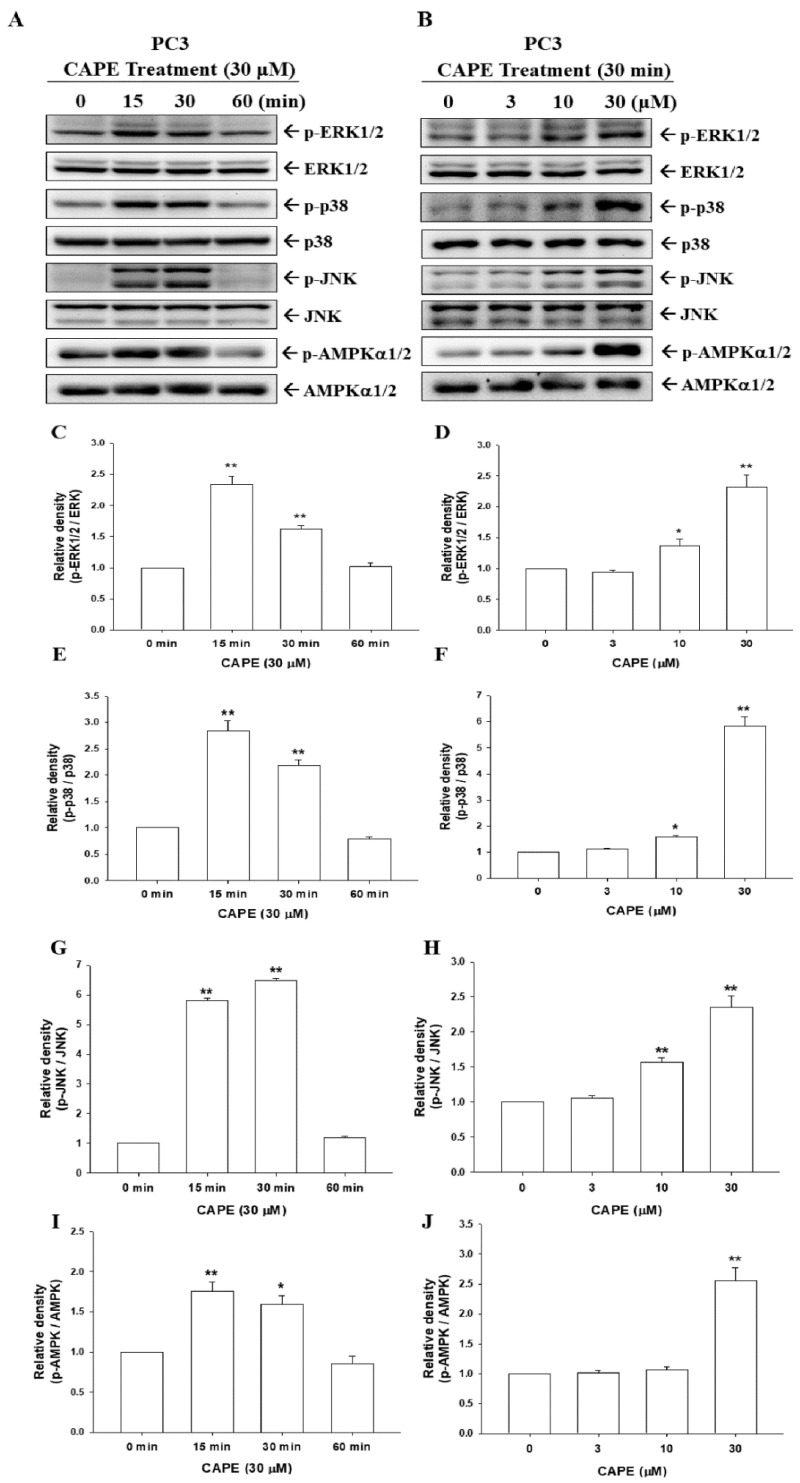
CAPE increases phosphorylation of ERK, p38, JNK, and AMPKα1/2 in PC-3 cells. (**A**) Time courses of ERK, p−ERK, p38, p−p38, JNK, p−JNK, AMPKα1/2, and p−AMPKα1/2 expressions were determined using immunoblot assays after various periods of 30 μM CAPE treatments in PC-3 cells. (**B**) Dose responses of ERK, p−ERK, p38, p−p38, JNK, p−JNK, AMPKα1/2, and p−AMPKα1/2 were determined using immunoblot assays after 30 min of various concentrations of CAPE treatments in PC-3 cells. The quantitative data are expressed as the intensity bands of the phosphorylation proteins relative to the total protein levels for ERK (**C**,**D**), p38 (**E**,**F**), JNK (**G**,**H**), and AMPK (**I**,**J**). * *p* < 0.05, ** *p* < 0.01.

**Figure 6 cancers-14-00274-f006:**
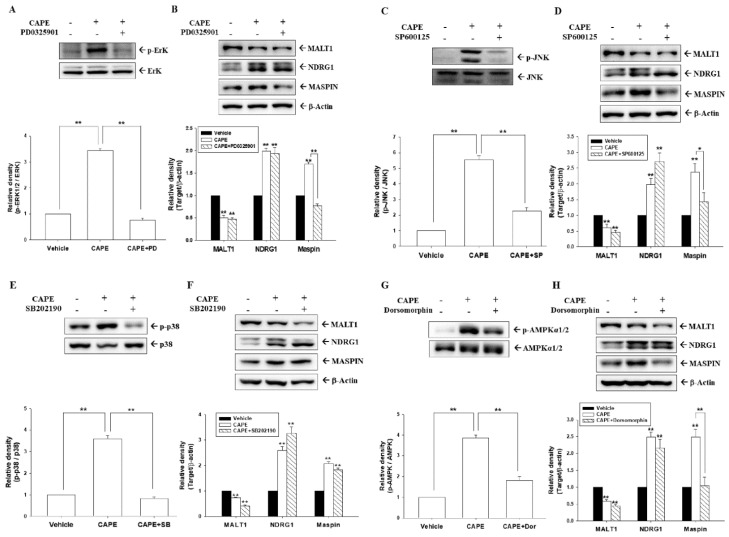
Inhibitors of MAPK and AMPKα1/2 modulate the expression of MALT1, NDRG1, and maspin in prostate carcinoma cells. The expressions of ERK and p−ERK (**A**), JNK and p−JNK (**C**), p38 and p−p38 (**E**), and AMPKα1/2 and p−AMPKα1/2 (**G**) were determined using immunoblot assays after 20 min of 30 μM CAPE treatments with (+) or without (−) pretreatment with the indicated concentrations of inhibitors for 1 h in PC-3 cells. The quantitative data are expressed as the intensity of protein bands of the phosphorylation proteins/total protein levels relative to the control solvent-treated group. The protein levels of MALT1, NDRG1, maspin, and β-actin in PC-3 cells after CAPE treatment with (+) or without (−) pretreatment with PD0325901 (**B**), SP600125 (**D**), SB202190 (**F**), or dorsomorphin (**H**). The quantitative data are expressed as the intensity bands of the phosphorylation proteins relative to the total protein levels or intensity of protein bands from the target genes/β-actin relative to the control solvent-treated group. * *p* < 0.05, ** *p* < 0.01.

**Figure 7 cancers-14-00274-f007:**
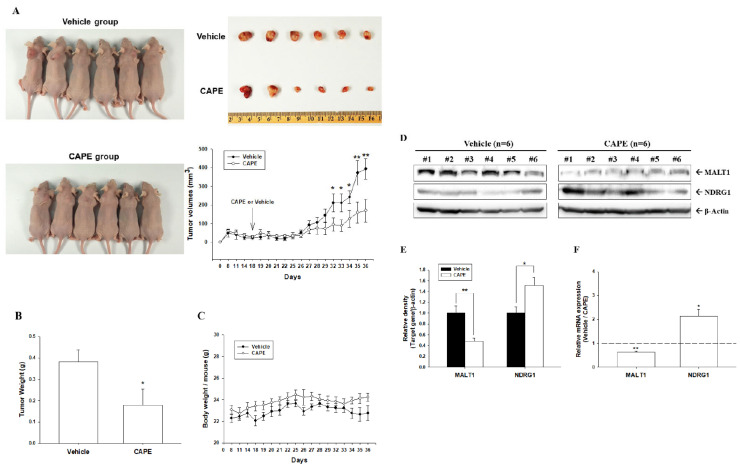
CAPE inhibits tumorigenesis of PC-3 cells in a xenograft mouse model. Athymic male nude mice were subcutaneously injected with PC-3 cells and randomly divided into two groups. When tumor volumes reached around 50 mm^3^ (day 18) after subcutaneous xenografting of PC-3 cells, nude mice received vehicle (0.1% DMSO in PBS; *n* = 6) or CAPE (10 mg/kg; *n* = 6) injected intraperitoneally once per day, 5 d/wk. (**A**) Tumor sizes were measured in the vehicle-treated (●) or CAPE-treated (○) groups. Mice were sacrificed on the 36th day after treatment initiation, and the tumors were excised. Photograph of the representative xenografted mice and tumors. (**B**) The quantitative data of tumor weight are presented as mean (±SE) of tumor weight in g. (**C**) The average body weight (mean ± SE) of mice during the experimental period. (**D**) Whole cell lysates of tumor samples from vehicle- or CAPE-treated groups were subjected to immunoblotting (**D**), and the quantitative data of immunoblot assays (**E**) are presented as mean fold induction of the protein levels of MALT1 and NDRG1 (±SE, *n* = 6) compared with the vehicle-treated group. (**F**) The quantitative data (±SE; *n* = 3) of RT-qPCR assays compared with the vehicle-treated group. * *p* < 0.05, ** *p* < 0.01.

## Data Availability

The data used to support the findings of this study are available from the corresponding author upon request.

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
