# Peer review of "The Antitumor Effect of Caffeic Acid Phenethyl Ester by Downregulating Mucosa-Associated Lymphoid Tissue 1 via AR/p53/NF-κB Signaling in Prostate Carcinoma Cells"

_cancers, 2022, doi:10.3390/cancers14020274_

Round 1
Reviewer 1 Report
The authors provide a detailed study on the effect of caffeic acid on phenethyl ester (CAPE) on prostate cancer.
Methodology relies on basic methodology including cell based assays and xenografts.
The main finding is that the CAPE attenuated androgen-induced PSA via the 439 AR, MALT1/NF-κB, or p53 signaling pathways in androgen-dependent and androgen-440 independent prostate carcinoma cells.
The manuscript is well-written and likely to be of interest to the readers of the journal.
Comments:
- What is the the concentration of CAPE used here in the cell culture and xenograft assays in respect to possible serum concentrations in humans? Are concentrations physiological or toxicological? Any idea from the literature how this would translate to concentrations achievable in patients? This should be further emphasized in the discussion.
- It would be of interest to evaluate the effect of CAPE in comparison to known inhibitors of AR axis already in clinical use (older and new antiandrogens).
- Would more detailed transcriptomics/epigenomic profiling with techniques currently available be of any value. Perhaps a though for this should be given in the discussion, how to move forvard?
- What would be your clinical intended use and study design if you were given free hands and resources to design such a study?
Reviewer 2 Report
The authors have represented a comprehensive study on a caffeic acid ester analog (CAPE) against prostate cancer. Their findings showed that CAPE exerts antitumor activity in a AR independent manner by attenuating the MALT1 inducing signaling pathway. It regulates MALT1 expression through P53 and NFkB.
The experimental design is very good and it has been nicely executed in most of the area.
Though Prostate cancer is new in the context, but similar study has been published in breast cancer, the authors must highlight the novelty of the manuscript in the abstract.
Fig. 7a, It is not clear, why until day 26, the tumor in the untreated animals were not growing, this is very unusual in subcutaneous cell derived xenograft.
The drug candidate, is only lowering the tumor progression rate, in that case the author must include a future perspective note in their discussion.
Fig. 7C, the authors must edit the Y axis, to focus in between 20-30g.
The immunoblots in Fig. 1A,1B and 7D are not clear enough. The authors should produce high resolution image.
It will very attractive to the reader to get an insight about the cellular uptake of CAPE in the tumor tissue.
Reviewer 3 Report
The authors present a series of experiments that describe a role for Caffeic acid phenethyl ester (CAPE) in the regulation of prostate cancer cell line growth in vitro and in a limited in vivo experiment. This work follows similar studies in cell lines from other tumor types by this group. While the experiments seem basically sound the question of how applicable they are to real prostate cancer in patients (or indeed to any other cancer) remains open. Discussion in this area is somewhat overstated.
Introduction (line 55) The authors say that CAFÉ inhibits the progression of several cancers. However, the papers cited, refs 7-13, refer to studies that were mostly in vitro with some in vivo xenograft work. This should be clearly acknowledged. Any number of compounds will slow growth in vitro, clinical application is an entirely different question.
The sentence that runs from line 66-70 “To date, …. carcinoma cells” contains a major non sequitur at the beginning of the second clause and should be broken up and edited for clarity.
Section 3.4 – the first sentence of this section is garbled and should be recast to make sense.
Section 3.7 First sentence – I don’t think “antigrowth” is a word. “the growth inhibitory effects of CAPE..” perhaps.
Discussion – the first paragraph of the discussion essentially repeats the overstatement problems identified in the introduction. This language should be more honest and speculative, putting the work in context is appropriate, overstating significance without acknowledging limitations is not.
Second paragraph – serum PSA is a marker of prostate pathologies, including cancer. For example, serum PSA can also be elevated in BPH. However, this manuscript is dealing with the cellular expression of the protein, not with the levels in serum, which reflect disruption of glandular architecture associated with pathologic growth, infection or organ trauma. In vitro, cells that express PSA do so directly in response to androgenic stimulation (rather than as a function of both differentiation status and androgenic stimulation as occurs in vivo). As such one would expect a direct link between reduced AR and PSA expression. This does not realyy reflect any change in the malignant nature of the cells. As the authors note there are links between NF-kB activation and PSA, that might be in response to the regulation of constitutively active AR splice variants by NF-kB (Austin et al papers). However, that is something of a side issue here.
Line 385 – should say “Consistent with this” or something similar “consistently” has a different meaning and from context doesn’t seem to be what the authors are striving for.
Lines 386-388 “CAPE attenuated androgen-induced PSA 386 via AR, MALT1/NF-κB, or p53 signaling pathways in androgen-dependent and -inde-387 pendent PC-3 cells” No data on PSA expression in PC3 cells is presented. And I would be deeply suspicious if any were. Please clarify.
Figures
Most of the gels in many of the figures (see for example figures 1 a,b and c) have been stretched along the x axis, for no apparent reason. This simply makes the gels, and associated text, appear unduly modified and distorted. In contrast gels in figure 6 have not been stretched and look much better. Ditto for many of the graphs presented.
A similar stretch has been applied to the mouse and tumor illustrations in figure 7, leading to bizarre appearing mice and stretched out tumors.
Overall this manuscript is well written and is easy to understand, although a critical review and judicious editing might smooth the language in a few places.
Finally, the title, while descriptive, is very long. The authors might consider something a little more succinct.
Round 2
Reviewer 2 Report
The authors have tried their best to improve the manuscript, following reviewers' suggestion. It can be accepted for publication.